# Identifying important factors for successful surgery in patients with lateral temporal lobe epilepsy

**Jae Rim Kim**[1], **Hyunjin Jo**[1¤], **Boram Park**[2], **Yu Hyun Park**[1,3,4], **Yeon Hak Chung**[1], **Young-Min Shon**[1,3], **Dae-Won Seo**[1], **Seung Bong Hong**[1], **Seung-Chyul Hong**[5], **Sang Won Seo**[1,2,3,4]*, **Eun Yeon Joo**[1]*

1 Department of Neurology, Neuroscience Center, Samsung Medical Center, Sungkyunkwan University School of Medicine, Seoul, South Korea, 2 Biomedical Statistics Center, Research Institute for Future Medicine, Samsung Medical Center, Seoul, South Korea, 3 Department of Health Sciences and Technology, SAIHST, Sungkyunkwan University, Seoul, South Korea, 4 Department of Intelligent Precision Healthcare Convergence, Sungkyunkwan University, Suwon, South Korea, 5 Department of Neurosurgery, Samsung Medical Center, Sungkyunkwan University School of Medicine, Seoul, South Korea

¤ Current address: Mark and Mary Stevens Neuroimaging and Informatics Institute, Keck School of Medicine, University of Southern California, Los Angeles, CA, United States of America
* sangwonseo@empal.com (SWS); ejoo@skku.edu (EYJ)

**Data Availability Statement:** All relevant data are within the paper and its Supporting Information files.

## Abstract

### Objective

Lateral temporal lobe epilepsy (LTLE) has been diagnosed in only a small number of patients; therefore, its surgical outcome is not as well-known as that of mesial temporal lobe epilepsy. We aimed to evaluate the long-term (5 years) and short-term (2 years) surgical outcomes and identify possible prognostic factors in patients with LTLE.

### Methods

This retrospective cohort study was conducted between January 1995 and December 2018 among patients who underwent resective surgery in a university-affiliated hospital. Patients were classified as LTLE if ictal onset zone was in lateral temporal area. Surgical outcomes were evaluated at 2 and 5 years. We subdivided based on outcomes and compared clinical and neuroimaging data including cortical thickness between two groups.

### Results

Sixty-four patients were included in the study. The mean follow-up duration after the surgery was 8.4 years. Five years after surgery, 45 of the 63 (71.4%) patients achieved seizure freedom. Clinically and statistically significant prognostic factors for postsurgical outcomes were the duration of epilepsy before surgery and focal cortical dysplasia on postoperative histopathology at the 5-year follow-up. Optimal cut-off point for epilepsy duration was eight years after the seizure onset (odds ratio 4.375, *p*-value = 0.0214). Furthermore, we propose a model for predicting seizure outcomes 5 years after surgery using the receiver operating characteristic curve and nomogram (area under the curve = 0.733; 95% confidence interval,

**Funding:** This study was granted by Samsung Medical Center Grant (OTC1190671). We state that the funders had no role in study design, data collection and analysis, decision to publish, or preparation of the manuscript.

**Competing interests:** There are no restrictions on sharing of data and/or materials of our manuscript. The authors declare that the research was conducted in the absence of any commercial or financial relationships that could be construed as a potential conflict of interest. This does not alter our adherence to PLOS ONE policies on sharing data and materials.

0.588–0.879). Cortical thinning was observed in ipsilateral cingulate gyrus and contralateral parietal lobe in poor surgical group compared to good surgical group ($p$-value < 0.01, uncorrected).

## Conclusions

The identified predictors of unfavorable surgical outcomes may help in selecting optimal candidates and identifying the optimal timing for surgery among patients with LTLE. Additionally, cortical thinning was more extensive in the poor surgical group.

## Introduction

Epilepsy surgery is an effective therapeutic option for patients with drug-resistant epilepsy [1]. Temporal lobe epilepsy (TLE) is the most frequent type of epilepsy that is treated surgically [2]. Approximately 10% of patients with TLE have seizures arising from the lateral temporal area; this condition is defined as lateral TLE (LTLE) [3]. As LTLE is diagnosed in a small number of patients, its characteristics and surgical outcomes are less well known compared to those of mesial TLE (MTLE).

LTLE is reported to have less favorable surgical outcomes than MTLE [3–6]. The chances of being seizure-free after surgery are lower in patients with LTLE than in those with MTLE. Hence, it is important to appropriately identify individuals who need to undergo epilepsy surgery. The etiological findings on magnetic resonance imaging (MRI) or histopathology can be used to predict surgical outcomes in patients with LTLE [7, 8]. Lateralized or localized ictal scalp electroencephalography (EEG) patterns have also been reported as factors that predict good surgical outcomes [8]. However, the abovementioned studies had limitations, such as a short-term follow-up (21.9±14 months) or small numbers of patients (29 patients).

Presurgical neuroimaging studies have provided prognostic implication of epilepsy surgery [9–12].

Recently, cortical thickness analysis has been adopted in epilepsy field [13–17]. The degree and location of cortical thinning differed depending on the type of epilepsy. For example, in TLE, the parahippocampal gyrus was thinned and in frontal lobe epilepsy (FLE) it was thinner in a relatively wider range of cortex. In neocortical epilepsy with normal MRI findings, frontal and hemispheric cortical thickness asymmetries indicated prognostic implications with a high positive value [18]. These findings suggest that the cortical thinning patten may become a prognostic factor for surgical outcomes.

In this retrospective cohort study, we aimed to identify the long- and short-term surgical outcomes in patients with LTLE. We investigated the prognostic implications of relatively objective clinical, electroencephalographic, and neuroimaging factors on surgical outcomes. We also developed a model to predict the probability of poor postoperative outcomes in LTLE.

## Materials and methods

We retrospectively reviewed the records of 1,172 patients with refractory epilepsy who underwent epilepsy surgery between January 1995 and December 2018 at Samsung Medical Center in South Korea. Patients with drug-resistant epilepsy underwent a comprehensive presurgical evaluation comprising neurological examination, video electroencephalography (VEEG), and temporal lobe MRI. When possible, $^{18}$F fluorodeoxyglucose- proton emission tomography (FDG-PET) and ictal and interictal single-photon emission computed tomography (SPECT)

scans were conducted to localize epileptic foci. After scalp VEEG monitoring, invasive VEEG monitoring was performed in patients suspected of having LTLE based on presurgical diagnostics. Patients were diagnosed with LTLE if the seizure onset zone was in the lateral temporal area, not in mesial temporal structures or basal temporal area based on an invasive study. The seizure onset zone was defined as any paroxysmal ictal pattern that was distinct from background activity with clinical symptoms. Patients who were followed-up for less than two years were excluded (Fig 1). Informed written consent was obtained from all participants when they

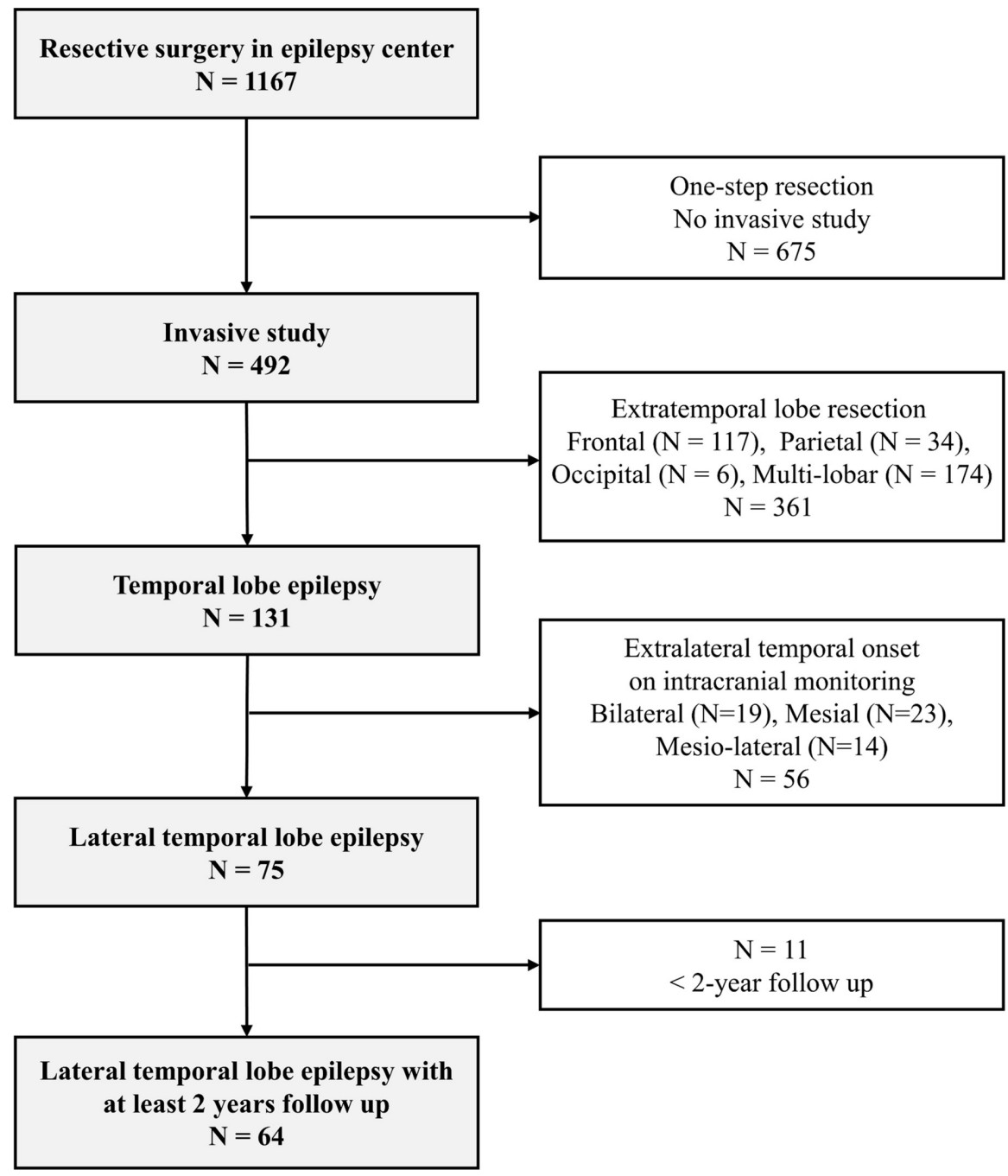

**Fig 1. Flow chart describing the selection process according to presurgical evaluation.**

admitted to epilepsy monitoring units (EMU), and it was obtained from the next of kin, care-takers, or guardians on behalf of the minors/children participants involved in this study. The Institutional Review Board of Samsung Medical Center approved the study protocol (IRB number 2021-03-068). We had not access to information that could identify individual participants during study.

## Demographic and clinical data

Clinical characteristics registered for each patient were recorded, including age at seizure onset, age at surgery, history of febrile seizures, major brain trauma, central nervous system infection or other medical diseases, and existence of auras or focal to bilateral tonic-clonic seizures.

## Scalp EEG

During scalp VEEG in EMU, the 10–10 system for scalp electrodes was used. Both interictal epileptiform discharges (INIEDs) and ictal epileptiform discharges (IEDs) were classified as localized (ipsilateral temporal) and non-localized (extratemporal or bilateral or none).

## Neuroimaging

MRI was performed using a GE Signa 1.5-Tesla scanner (GE Medical Systems, Inc., Milwaukee, WI, USA) or a 3.0-Tesla scanner (Philips, Best, The Netherlands). All studies included spoiled gradient echo, T2-weighted, and fluid-attenuated inversion recovery (FLAIR) imaging. The MRI results were classified into three subtypes: lateral temporal lesions, such as focal cortical dysplasia (FCD) or tumorous lesion; mesial temporal lesion but normal lateral cortex, such as hippocampal sclerosis (HS); and normal.

Additionally, we analyzed cortical thickness from some available MRI data. The cortical thickness was measured by the same measurement procedures has been described in the author's previous study [19].

FDG-PET was performed (GE Advance PET scanner, GE Medical Systems, Inc.) 4 or more hours after an intravenous injection of 7–10 mCi (259–370 MBq) of FDG. Hypometabolism was determined by visual assessment. FDG-PET results were classified as localized (ipsilateral temporal) and non-localized (extratemporal or bilateral or none).

Brain SPECT scans were performed 30–60 min after injection of 25 mCi 99mTc-ethyl cysteinate dimer using a three-headed Triad XLT system (Trionix Research Laboratory, Inc., Twinsburg, OH, USA). Subtraction Ictal SPECT Co-registered to MRI (SISCOM) was analyzed on an offline workstation using the Analyze 7.5 software (Biomedical Imaging Resource, Mayo Foundation, Rochester, MN, USA). SISCOM results were classified as localized (ipsilateral temporal) and non-localized (extratemporal or bilateral or none).

## Intracranial VEEG and surgery

All participants who were suspected LTLE underwent the invasive monitoring using a combination of subdural grids/strips with or without depth electrodes to confirm the ictal onset zone and determine the extent of resection. Once the ictal onset zone was confirmed using sufficient data, the extent of resection was determined. Resection included lesionectomy or corticectomy, which were done alongside the removal of intracranial electrodes. In some cases, anterior temporal lobectomy (ATL) or amygdalohippocampectomy (AH) was necessary.

Surgical specimens were reviewed for histopathological and immunohistochemical analysis. The classification of pathology was based on the International League Against Epilepsy guidelines for FCD [20] and HS [21].

## Surgical outcome

Postoperative seizure outcomes were classified based on Engel's classification [22]. Patients were divided into two outcome groups: good (corresponding to Engel's class I) and poor (corresponding to Engel's class II–IV) outcomes.

## Statistical analyses

The demographic and clinical characteristics of the patients are summarized as number (percentile) for categorical variables and mean ± standard deviation for continuous variables. To compare two groups, we applied the Chi-square or Fisher's exact test for categorical variables. Student's t-test or Mann-Whitney U test was performed for continuous variables. Logistic regression analysis was used to identify the independent risk factors for poor surgical outcomes. Variables with a $p$-value $< 0.1$ in the univariable model were included in a multivariable logistic regression analysis.

The receiver operating characteristic (ROC) curve analysis and the area under the curve (AUC) was used to measure how well the model discriminated between seizure-free and recurrent seizure patients. A nomogram was created based on multivariable logistic regression models to provide a visual representation of the model. The optimal cut-off value for the duration of epilepsy prior to surgery was determined using a logistic regression model that gives the maximum difference between good and poor surgical outcomes at five years. Statistical analyses were performed using Statistical Package for the Social Sciences version 27.0 (IBM Corp., Armonk, NY, USA). Statistical significance was set at $p$-value $< 0.05$.

For cortical thickness analysis of the MRI data, we used a MATLAB-based toolbox (freely available online at the University of Chicago website: http://galton.uchi cago.edu/faculty/ InMemoriam/worsley/research/surfstat/). To identify the cortical thinning pattern in the groups, we analysed localized differences in cortical thickness between the good surgical outcome and poor surgical outcome groups using a general linear model after controlling for age, sex, and ICV. To illustrate trends, significances were displayed at an uncorrected threshold of $p$-value $< 0.01$.

## Inclusivity in global research

Additional information regarding the ethical, cultural, and scientific considerations specific to inclusivity in global research is included in the Supporting Information (S1 File).

## Results

### Patients' characteristics

Sixty-four patients with LTLE were enrolled, and their demographic and clinical data were analyzed (Table 1). Among the participants, 62.5% were male, and the mean age at seizure onset was 18.5 years (standard deviation [SD]: 8.9 [range: 1–34]). The mean age at surgery was 29.0 years (SD, 11.5 [range, 9.0–61.0]), and the mean duration of epilepsy prior to surgery was 10.5 years (SD, 7.4 [range, 1.0–31.0]). 49 (76.6%) patients experienced focal to bilateral tonic-clonic seizures before surgery, and 26 (40.6%) patients experienced more than one seizure per week. There was no difference between the groups, except for the preoperative disease duration and follow up periods after surgery.

## Presurgical diagnostics

Table 2 details the pre-surgical evaluation results of the participants. Interictal EEG showed concordant with the resected lobe in 52 patients (81.3%). All patients had seizures recorded during scalp VEEG monitoring, and in 48 (75.0%) patients, ictal EEG showed concordance. There was no difference according to surgical outcomes.

Temporal lobe MRI was performed in all patients, and 47 (73.4%) patients showed lesions in the lateral temporal areas. Changes in the hippocampus or amygdala were observed in seven (10.9%) patients, but no abnormalities in the lateral temporal cortex were observed. While 10 patients (15.6%) had normal MRI findings. In 55 (85.9%), FDG-PET was performed, and 25 (39.1%) patients showed hypometabolism involved extratemporal areas. SISCOM with interictal and ictal SPECT was performed in 40 patients (62.5%), and five (6.3%) patients showed hyperperfusion areas beyond the temporal areas. There was a difference in structural imaging findings between two groups, but no difference in the functional imaging findings.

## Surgery, surgical outcome, and histopathology

All patients underwent resective surgery according to their respective invasive VEEG monitoring results. The mean follow-up duration after surgery was 8.4 years (SD, 6.3 [range, 2.0–26.0] years). All patients, except one, were followed up for more than 5 years. Out of the 64 patients, 39 (60.9%) patients became seizure-free two years after surgery. 45 of the 63 (71.4%) patients had become seizure-free five years after surgery.

The study also found that FCD was the most common underlying pathology, present in 26 (40.6%) of the patients. Glioneuronal tumor was the second most common pathology, present in 23 (35.9%) patients. Vascular malformation was present in 9 (14.1%) patients, and unspecific gliosis was present in 6 (9.4%) patients.

## Factors that predict seizure outcomes at 2 and 5 years after surgery

Table 3 shows the variables associated with seizure outcomes at the 2-year follow-up using the logistic regression analysis. Significant prognostic factors were duration of epilepsy prior to

**Table 1. Participants' demographics and clinical characteristics.**

|  | Good (I) | Poor (II-IV) | $p$-value |
|---|---|---|---|
|  | ($n$ = 39) | ($n$ = 25) |  |
| **Gender** |  |  |  |
| Male, $n$ (%) | 24 (61.5) | 16 (64.0) | 0.843 |
| Female, $n$ (%) | 15 (38.5) | 9 (36.0) |  |
| **Age at Seizure onset, year** | 18.3±9.4 | 18.8±8.1 | 0.839 |
| **Age at Surgery, year** | 27.2±12.8 | 31.8±8.6 | 0.057 |
| **Duration of epilepsy prior to surgery, year** | 8.9±7.0 | 13.0±7.6 | 0.034* |
| **Follow up after surgery, year** | 6.9±6.2 | 10.7±5.9 | 0.005* |
| **Side of surgery** |  |  |  |
| Left, $n$ (%) | 22 (56.4) | 20 (80) | 0.053 |
| Right, $n$ (%) | 17 (43.6) | 5 (20) |  |
| **Seizure frequency ($\geq$1/week)** | 18 (46.2) | 8 (32.0) | 0.261 |
| **FBTCS before surgery** | 28 (71.8) | 21 (84) | 0.261 |

Continuous variables are presented as mean ± standard deviation.

Categorical variables are presented as N (%).

FBTCS, Focal to bilateral tonic-clonic seizure.

*$p$-value < 0.05.

**Table 2. Participants' presurgical diagnostics.**

| | Good (I) | Poor (II-IV) | *p*-value |
|---|---|---|---|
| | (*n* = 39) | (*n* = 25) | |
| **Interictal EEG pattern** | | | |
| Localized, *n* (%) | 31 (79.5) | 21 (84.0) | 0.751 |
| Non-localized, *n* (%) | 8 (20.5) | 4 (16.0) | |
| Total | 39 | 25 | |
| **Ictal EEG pattern** | | | |
| Localized, *n* (%) | 27 (69.2) | 21 (84.0) | 0.183 |
| Non-localized, *n* (%) | 12 (30.8) | 4 (16.0) | |
| Total | 39 | 25 | |
| **MRI findings** | | | |
| Lateral lesion | 33 (84.6) | 14 (56.0) | 0.044* |
| Mesial lesion | 2 (5.1) | 5 (20.0) | |
| Normal | 4 (10.3) | 6 (24.0) | |
| Total | 39 | 25 | |
| **PET pattern** | | | |
| Localized, *n* (%) | 20 (51.3) | 10 (40.0) | 0.198 |
| Non-localized, *n* (%) | 12 (30.8) | 13 (52.0) | |
| Total | 32 | 23 | |
| **SISCOM pattern** | | | |
| Localized, *n* (%) | 24 (61.5) | 12 (48.0) | 0.539 |
| Non-localized, *n* (%) | 2 (5.1) | 2 (8.0) | |
| Total | 26 | 14 | |
| **Pathology of lateral temporal area** | | | |
| Focal cortical dysplasia, *n* (%) | 12 (30.8) | 14 (56) | 0.092 |
| Glioneuronal tumor, *n* (%) | 17 (43.6) | 6 (24.0) | |
| Vascular malformation, *n* (%) | 9 (23.1) | 0 (0) | |
| Other, *n* (%) | 1 (2.6) | 5 (20) | |
| Total | 39 | 25 | |

Categorical variables are presented as *N* (%).

*\*p*-value < 0.05.

**Table 3. Univariable and multivariable logistic regression for postsurgical outcome two years after surgery.**

| | Univariable OR | *p*-value | Multivariable OR | *p*-value |
|---|---|---|---|---|
| | OR (95% CI) | | OR (95% CI) | |
| **Duration of epilepsy prior to surgery, year** | 1.080 (1.004–1.161) | 0.038* | 1.057 (0.98–1.141) | 0.150 |
| **Interictal EEG (non-localized)** | 0.738 (0.197–2.768) | 0.653 | | |
| **Ictal EEG (non-localized)** | 0.429 (0.121–1.522) | 0.190 | | |
| **MRI findings** | | | | |
| Lateral lesion (ref) | 1 | | 1 | |
| vs Mesial lesion | 5.893 (1.019–34.079) | 0.048* | 3.959 (0.623–25.182) | 0.162 |
| vs Normal | 3.536 (0.862–14.499) | 0.079 | 2.277 (0.45–11.523) | 0.316 |
| **PET pattern (non-localized)** | 2.167 (0.727–6.455) | 0.165 | | |
| **Histopathology (FCD)** | 2.864 (1.01–8.119) | 0.048* | 1.514 (0.421–5.441) | 0.585 |

OR, odds ratio; CI, confidence interval; EEG, electroencephalography; MRI, magnetic resonance imaging; PET, positron emission tomography; FCD, focal cortical dysplasia.

*\*p*-value < 0.05.

surgery, potentially epileptogenic lesions in the mesial but normal lateral temporal structures on preoperative MRI, and FCD on histopathology in the univariable analysis. However, no independent prognostic factors were identified in multivariable analysis.

Table 4 shows the variables associated with seizure outcomes at the 5-year follow-up. Scalp EEG or imaging factors did not affect seizure outcomes, as determined by univariable analysis. However, FCD on postoperative histopathology was found to be the only significant risk factor for seizure outcome (odds ratio [OR], 4.92; 95% confidence interval [CI], 1.52–15.91) in both univariable and multivariable analysis. The duration of epilepsy before surgery was found to affect seizure outcomes, but the difference was not statistically significant. To increase statistical significance, we attempted to find the optimal cut-off point for preoperative epilepsy duration. The cut-off value was determined to be eight years after the onset (OR 4.375, $p$-value = 0.0214). However, the analysis of the duration of epilepsy as a categorical variable was only a significant prognostic factor in the univariable analysis and not in the multivariable analysis (Table 4(B)).

Furthermore, we suggested a model for predicting outcomes at 5 years after surgery using a ROC curve and nomogram. The duration of epilepsy prior to surgery showed a clear trend towards association with poorer outcomes, although this was not statistically significant due to the small group size. We included the duration of epilepsy in our model, despite its insignificant $p$-value, because we believed that with a larger sample size, the variable may become

**Table 4. Univariable and multivariable logistic regression for postsurgical outcome five years after surgery.**

(A) Duration of epilepsy prior to surgery included as a continuous variable

|  | Univariable OR OR (95% CI) | $p$-value | Multivariable OR OR (95% CI) | $p$-value |
|---|---|---|---|---|
| **Duration of epilepsy prior to surgery, year** | 1.07 (1.00–1.16) | 0.064 | 1.06 (0.97–1.15) | 0.194 |
| **Interictal EEG (non-localized)** | 0.80 (0.19–3.37) | 0.761 | | |
| **Ictal EEG (non-localized)** | 0.31 (0.06–1.53) | 0.150 | | |
| **MRI findings** | | | | |
| Lateral lesion (ref) | 1 | | | |
| vs Mesial lesion | 2.39 (0.46–12.34) | 0.299 | | |
| vs Normal | 2.12 (0.51–8.91) | 0.304 | | |
| **PET pattern (non-localized)** | 1.24 (0.38–3.98) | 0.723 | | |
| **Histopathology (FCD)** | 4.92 (1.52–15.91) | 0.007* | 4.20 (1.26–13.94) | 0.019* |

(B) Duration of epilepsy prior to surgery included as a categorical variable

|  | Univariable OR OR (95% CI) | $p$-value | Multivariable OR OR (95% CI) | $p$-value |
|---|---|---|---|---|
| **Duration of epilepsy prior to surgery (>8 year)** | 4.38 (1.24–15.38) | 0.021* | 3.23 (0.87–12.06) | 0.081 |
| **Interictal EEG (non-localized)** | 0.80 (0.19–3.37) | 0.761 | | |
| **Ictal EEG (non-localized)** | 0.31 (0.06–1.53) | *0.150* | | |
| **MRI findings** | | | | |
| Lateral lesion (ref) | 1 | | | |
| vs Mesial lesion | 2.39 (0.46–12.34) | 0.299 | | |
| vs Normal | 2.12 (0.51–8.91) | 0.304 | | |
| **PET pattern (non-localized)** | 1.24 (0.38–3.98) | 0.723 | | |
| **Histopathology (FCD)** | 4.92 (1.52–15.91) | 0.007* | 3.83 (1.13–12.95) | 0.031* |

OR, odds ratio; CI, confidence interval; EEG, electroencephalography; MRI, magnetic resonance imaging; PET, positron emission tomography; FCD, focal cortical dysplasia.

*$p$-value < 0.05.

statistically significant and provide useful information for predicting seizure outcomes after surgery. According to the ROC curve analysis, the duration of epilepsy prior to surgery and histopathological results could be used as predictors of seizure outcomes at 5 years after surgery (AUC = 0.733; 95% CI, 0.588–0.879). Similar results were obtained when the duration of epilepsy was used as a categorical variable (AUC = 0.736; 95% CI, 0.595–0.878). The nomogram for the model is shown in Fig 2.

## Cortical thickness analysis predicting seizure outcomes at 5 years after surgery

Cortical thickness analysis was conducted on 46 patients, and data from LTLE were combined and analyzed relative to the epileptogenic lobe to increase statistical power. In this analysis, the left hemisphere means ipsilateral and the right hemisphere means contralateral.

The results are displayed in Fig 3. A comparison between the poor and good outcome groups showed that there was thinning in the ipsilateral midcingulate gyrus and contralateral superior parietal lobule (p < 0.01, uncorrected).

## Subgroup analysis in mesial temporal lesion

Seven patients in the study did not have any lesions in the lateral temporal cortex but had changes in the mesial temporal region. Among these patients, six showed hippocampal signal change or atrophy, while one patient had amygdalar signal change. Two patients had a history of encephalitis, and one patient had a history of head trauma.

Six patients underwent AH. Three patients had total AH, while three had partial AH. Histological examination showed HS in three patients and normal in three others of six mesial temporal specimens. FCD was found in five of seven lateral temporal specimens. Microdysgenesis was found in another, and subpial gliosis was found in the other specimen.

Two out of seven patients were seizure-free at the 2-years follow-up, while four were seizure-free at the 5-years follow-up. Two patients who were seizure-free at 2-year follow-up remained seizure-free at the 5-year follow-up. All the seizure-free patients underwent total AH, while all patients who underwent partial AH had poor surgical outcomes.

## Discussion

In this study, we determined postsurgical outcomes and verified possible prognostic factors in patients with LTLE during the long-term follow-up period. We found that 60.9% of patients became seizure-free during the short-term follow-up period. In the long-term follow-up period, the rate of achieving seizure freedom improved to 71.4%. Our findings highlighted that FCD on histopathology and the duration of epilepsy prior to surgery are important prognostic factors for long-term surgical outcomes in patients with LTLE. The optimal cut-off value was 8 years after seizure onset. The nomogram we developed can help clinicians provide individualized predictions of surgical outcomes for their patients. The cortical thickness analysis we conducted also provides insights into more marked atrophy with poor surgical outcomes.

Several previous studies have selectively investigated surgical prognostic factors of LTLE, but they have some limitations. For example, one study included a short follow-up period of only one month of follow-up [7], while other studies had a small number of participants [8, 23]. However, our study has the advantage of a relatively long follow-up period of 8.2 years on average and a relatively large number of participants. Our findings of surgical outcomes are consistent with previous studies that reported rate of 60–79% [7, 8, 23]. Interestingly, our study showed surgical outcome improved over time. This may be related to study population. As reported by Tellez-Zenteno [24], long-term seizure freedom was highest in patients with

**(A)**

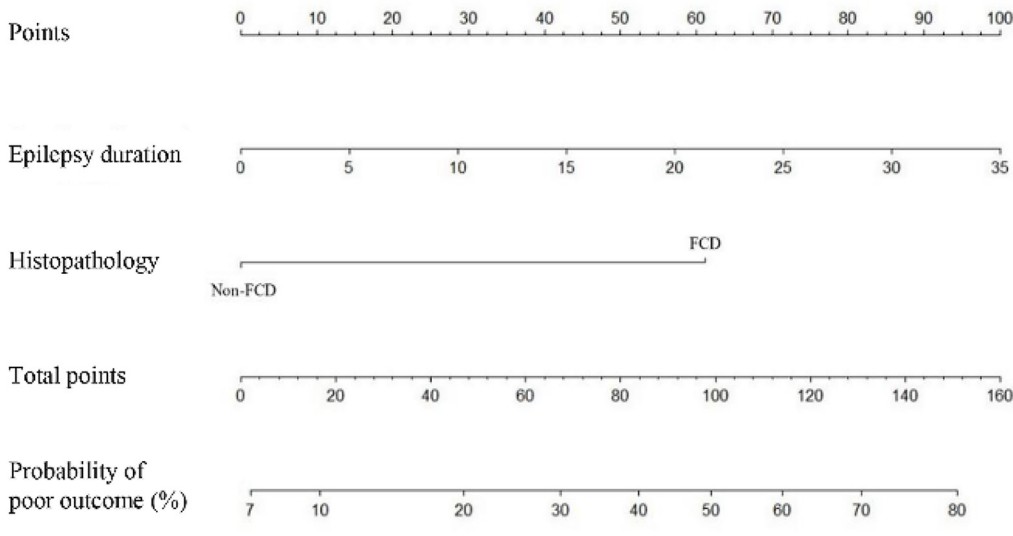

**(B)**

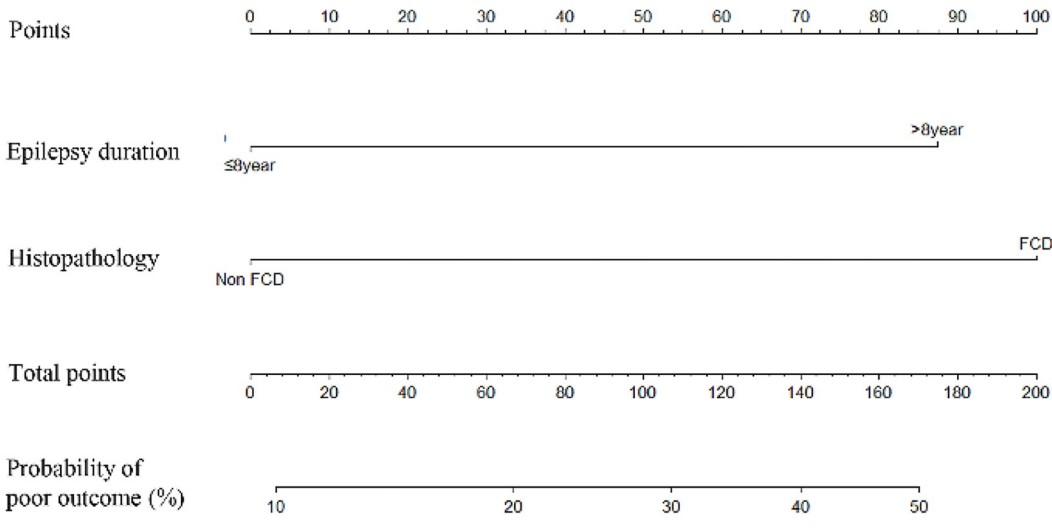

**Fig 2. Prognostic nomogram to predict the surgical outcome 5 years after surgery.** To use the nomogram, locate the patient's position on the scale associated with each predictor. The top axis displays prognostic points. Connect the position on each variable axis to the Points (top) axis to determine the number of points corresponding to the appropriate variable position. Calculate the total points for all variables, and then determine the appropriate position on the total points axis and connect it with the associated position on the probability of poor outcome (bottom) to determine the patient's individual risk. This nomogram is presented in two forms: (A) Duration of epilepsy prior to surgery included as a continuous variable, and (B) Duration of epilepsy prior to surgery included as a categorical variable.

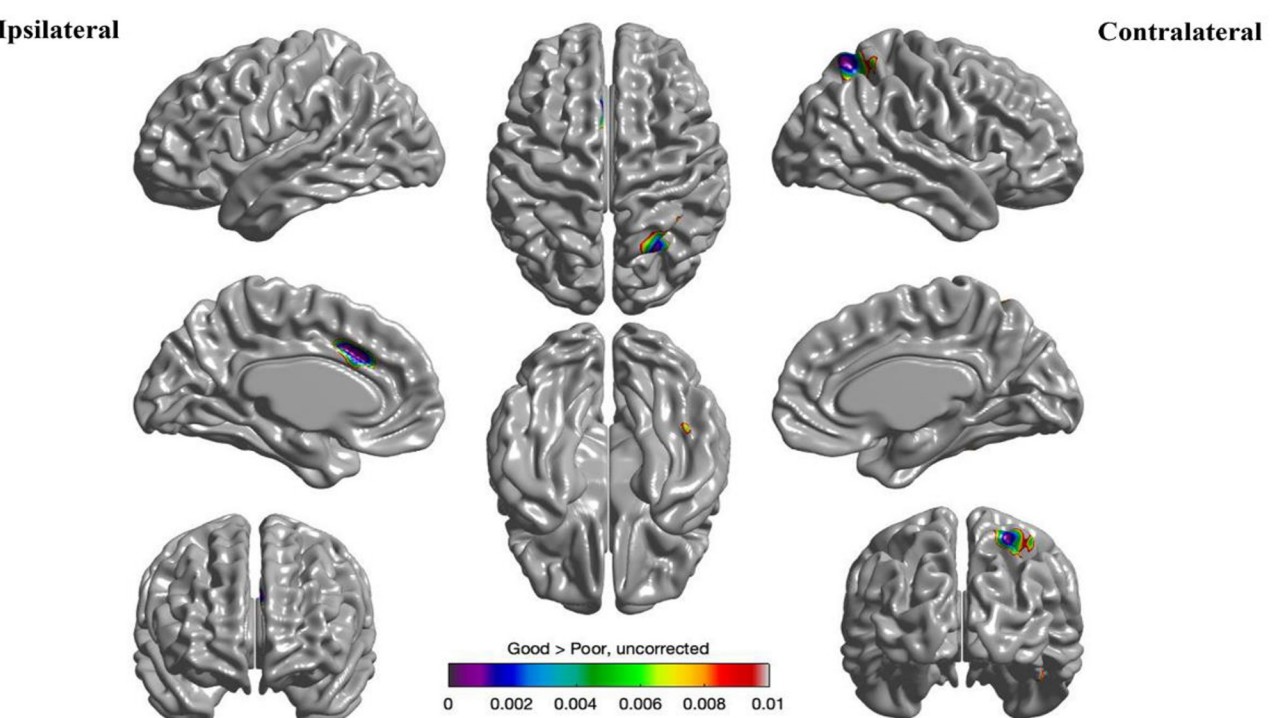

**Fig 3. Predictors of postsurgical outcome five years after surgery.** This shows the areas of cortical thinning in patients who experienced residual seizures (Engel's Class II–IV) compared to those who were seizure-free (Engel's Class I) after surgery. The areas of significant cortical thinning are displayed with uncorrected *p*-value less than 0.01.

tumor, but patients older than 50 years at time of surgery was lowest compared to short-term seizure freedom in temporal lobe surgery. We included patients with tumorous lesions and the average age of surgery was 29.0 years. There are also results that surgical outcomes improved patients with FCD [25], and it is agreement with our results.

Our study also showed that FCD was associated with poor surgical outcomes, consistent with other studies [8], while tumors on MRI or histopathology were associated with good surgical outcomes [7, 8]. We found that lesions on MRI showed a trend toward better surgical outcomes in univariable analysis. In contrast to some previous studies [8], our study did not identify a prognostic implication for the ictal EEG pattern.

FCD is a common cause of intractable epilepsy, characterized by a spectrum of regional cortical malformations [20]. According to our study, FCD identified through histopathology was the most reliable predictor of poor surgical outcomes in patients with LTLE. A previous study of patients with FCD conducted in the same epilepsy center reported that 61.7% and 39.5% of patients were seizure-free at 2 and 5 years after surgery, respectively [26]. The long-term surgical outcome for patients with FCD was worse than that of patients with LTLE. Although there were the differences between groups, including various epileptogenic foci, this finding supports the notion that FCD is a prognostic factor for worse outcomes.

However, the predictive value of FCD has been controversial, which could be attributed to differences in the study population [8, 27]. Some studies have suggested that complete resection, including dysplasia, is more predictable than FCD itself [28, 29]. This may be one of the reasons why FCD attributes to worse prognosis in our study. In patients with non-lesional MRI, resection margin was determined only by ictal onset zone. It might lead to incomplete resection and seizure recurrence. All of them were confirmed FCD on histopathology.

This study showed that the duration of epilepsy prior to surgery may have an impact on surgical outcomes, especially for patients with LTLE. Previous studies, including meta-analyses, has suggested that patients with shorter epilepsy duration have better surgical outcomes, regardless of the type of epilepsy [28, 30, 31]. A systematic review has also suggested that patients who are appropriate candidates for surgery should be referred earlier. However, the evidence for this recommendation has been weak due to group heterogeneity. Our results support the idea that early referral to an epilepsy surgery center is beneficial for patients with LTLE. Remarkably, there was a significant difference in surgical outcomes 8 years after the onset of epilepsy, with the probability of worse outcome being 4.4 times higher in the longer duration group than in the shorter duration group. Therefore, it is recommended that patients with LTLE are referral to an epilepsy surgery center within 8 years of onset. However, other studies have shown that the duration of epilepsy is not always associated with surgical outcomes [4, 32, 33], indicating that this recommendation should be applied to a specific population.

To predict surgical outcomes for patients with epilepsy, several studies have developed models that utilize various factors such as seizure frequency, history of generalized tonic-clonic seizures, MRI, and epilepsy duration [34]. Nomograms, which are graphical tools to predict outcomes, have also been used to predict surgical outcome based on various factors such as sex, seizure frequency, secondary seizure generalization, type of surgery, pathologic cause, age at epilepsy onset, age at surgery, epilepsy duration at the time of surgery, and surgical side [35]. However, many of these studies have included heterogeneous epilepsy groups, which has led to uncertainty about the accuracy of these models. A consecutive study disputed that these models are not superior to clinical judgment and have stressed the need for better tools [36]. Thus, our study proposes a new nomogram for predicting surgical outcomes in patients with LTLE based on epilepsy duration and histopathology. However, external validation of this nomogram in other patient cohorts is necessary to assess its accuracy and generalizability.

Cortical thickness has been studied in relation to surgical outcomes, with distinct patterns of cortical thinning associated with different epilepsy syndromes [17]. In TLE, the poor surgical outcome group exhibited more pronounced ipsilateral temporal, contralateral parietal and occipital, and bilateral insular thinning compared to the good surgical outcome group [13]. Meanwhile, neocortical epilepsy is associated with cortical thinning in the ipsilateral hemisphere, particularly in the frontal lobe, which leads to poor surgical outcomes [18]. For FLE, cortical volume loss in frontal and extrafrontal regions is a predictor of seizure outcomes [10]. In other words, if there are structural alterations outside of the resected lobe, regardless of the epilepsy types, the surgical prognosis is unfavorable. This study found that the poor surgical outcome group had more pronounced ipsilateral midcingulate and contralateral parietal thinning than the good surgical outcome group, indicating epileptic networks are wider in patients with residual seizures. Other modalities, such as FDG-PET and stereo EEG, have also shown similar results in worse outcome group, with similar extratemporal hypometabolism [11, 37] and early extratemporal propagation in stereo EEG [38].

In addition, we conducted a review of seven patients who had lesions in the mesial temporal area, with six of them undergoing resection surgery that included AH. The results showed that the patients who underwent partial resection of the hippocampus experienced seizure recurrence, while those who had total resection were seizure-free during long-term follow-up. Given that our findings are based on a limited number, the results from them suggest that partial resection of the hippocampus may lead to poor surgical outcomes. Previous studies have indicated that the large extent of hippocampus resection is associated with better surgical outcomes [33, 39]. However, other studies have shown that the extent of resection is not linked to seizure outcomes [40, 41], and a larger resection may lead to a higher risk of memory decline

[39, 40] As a result, the resection of the mesial temporal structure should be individualized for each patient with LTLE to minimize the risk of postoperative seizures and memory impairment.

Although the subgroup of patients who had no lesion in the lateral temporal cortex based on MRI, pathologic results showed FCD in all patients, except for two. The patients who did not have FCD underwent surgery before 2000, when the FCD classification was not yet been proposed [42]. If the specimens were reviewed now, it is likely that they would be classified as FCD.

As confirmed by invasive monitoring, ictal onset zones were in the lateral temporal area in all patients including the subgroup. As a result, signal changes observed in the mesial temporal area could be attributed to other mechanisms. First, seizures can induce signal changes in MRI, which are transient abnormalities that occur after seizure activity in the epileptogenic zone or distinct regions connected by an epileptic network [43]. In some cases, MRI scans were performed after clustering seizures on the last day of EMU monitoring, which could impact the observed signal changes on MRI. Second, the kindling model might be related to these alterations. Kindling refers to a change in seizure characteristics and behaviors resulting from recurrent seizures due to focal electrical stimulation [44]. Limbic circuits, including the hippocampus and amygdala, are highly vulnerable to kindling, which can eventually lead to epileptogenicity in affected regions. One study showed changes observed in the mesial temporal area may be due to the kindling process [45].

A previous study conducted at the same hospital on patients with MTLE found that 85.5% and 83.5% of patients were seizure free in 2 and 5 years after surgery, respectively [33]. This suggests that patients with MTLE have better surgical outcomes compared to those with LTLE, which is consistent with the findings of other studies [3–6]. Dolezalova et al. reported that 69% of patients with MTLE were seizure-free after one year, whereas only 42% of patients with LTLE were seizure-free after surgery [5]. Similarly, Lee et al. found that 70% of patients with MTLE were seizure-free, while only 33% of patients with LTLE achieved seizure-freedom [6]. However, one study showed no significant difference in seizure-free outcomes between the MTLE and LTLE groups [46]. In patients with MTLE, bitemporal INIED, MRI findings, and bitemporal hypometabolism on FDG-PET were associated with poorer surgical outcomes [33]. However, the predictive value of EEG, MRI, and PET in postsurgical outcomes was not identified in patients with LTLE.

This study has several limitations that should be considered when interpreting the results. Firstly, as a retrospective study, there is a potential for incomplete data and bias due to unmeasured confounders. However, to minimized this, the study used a standardized set of pre-surgical evaluations for all patients. Secondly, the study included a small number of patients from a single epilepsy center, which limits the generalizability of the findings to all patients with LTLE. Future studies with larger sample sizes and multiple centers are needed to confirm the results. Thirdly, the nomogram developed in this study needs to be applied to an external validation cohort and calibration before its clinical utility can be established. Fourthly, the absence of healthy controls in the cortical thickness analysis is another limitation. Additionally, factors such as the side of TLE and duration of epilepsy may influence cortical thickness but were not analyzed separately due to small sizes. Therefore, further investigations are needed to address these limitations and confirm the findings.

In conclusion, this study indicates that nearly 70% of the patients with LTLE may achieve long-term seizure freedom through resective surgery. However, FCD identified on histopathology was found to be a predictor of poor surgical outcomes. The study suggests that resective surgery should be considered within 8 years of epilepsy onset. We also developed a nomogram based on FCD and duration of epilepsy before surgery, which could help select

suitable candidates. If confirmed prospectively, these factors can help select optimal candidates and identify the optimal timing for surgery in patients with LTLE. Additionally, the study found that cortical thinning was more extensive in the group with poor surgical outcomes.

## Supporting information

**S1 File. Questionnaire on inclusivity in global research.**
(DOCX)

**S2 File. Data set.**
(XLSX)

## Author Contributions

**Conceptualization:** Jae Rim Kim, Sang Won Seo, Eun Yeon Joo.

**Data curation:** Jae Rim Kim, Boram Park, Yu Hyun Park, Yeon Hak Chung, Young-Min Shon, Dae-Won Seo, Seung Bong Hong, Seung-Chyul Hong, Eun Yeon Joo.

**Formal analysis:** Jae Rim Kim, Hyunjin Jo, Boram Park, Yu Hyun Park, Yeon Hak Chung, Sang Won Seo.

**Investigation:** Jae Rim Kim, Hyunjin Jo.

**Methodology:** Jae Rim Kim, Boram Park, Yu Hyun Park, Sang Won Seo.

**Resources:** Young-Min Shon, Dae-Won Seo, Seung Bong Hong, Seung-Chyul Hong, Eun Yeon Joo.

**Software:** Yu Hyun Park.

**Supervision:** Young-Min Shon, Dae-Won Seo, Seung Bong Hong, Seung-Chyul Hong, Sang Won Seo, Eun Yeon Joo.

**Validation:** Yu Hyun Park.

**Visualization:** Boram Park, Yu Hyun Park.

**Writing – original draft:** Jae Rim Kim, Boram Park, Yu Hyun Park, Yeon Hak Chung.

**Writing – review & editing:** Jae Rim Kim, Hyunjin Jo, Sang Won Seo, Eun Yeon Joo.

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
