## [Decision Letter · Decision Letter 0]

12 Apr 2023

PONE-D-23-07317Identifying Important Factors for Successful Surgery in Patients with Lateral Temporal Lobe EpilepsyPLOS ONE

Dear Dr. Joo,

Thank you for submitting your manuscript to PLOS ONE. After careful consideration, we feel that it has merit but does not fully meet PLOS ONE’s publication criteria as it currently stands. Therefore, we invite you to submit a revised version of the manuscript that addresses the points raised during the review process.

We look forward to receiving your revised manuscript.

Kind regards,

Tommaso Martino, M.D.

Academic Editor

PLOS ONE

2. 1)Please include a complete copy of PLOS’ questionnaire on inclusivity in global research in your revised manuscript. Our policy for research in this area aims to improve transparency in the reporting of research performed outside of researchers’ own country or community. The policy applies to researchers who have travelled to a different country to conduct research, research with Indigenous populations or their lands, and research on cultural artefacts. The questionnaire can also be requested at the journal’s discretion for any other submissions, even if these conditions are not met.  Please find more information on the policy and a link to download a blank copy of the questionnaire here: https://journals.plos.org/plosone/s/best-practices-in-research-reporting. Please upload a completed version of your questionnaire as Supporting Information when you resubmit your manuscript.

2) We noticed you have some minor occurrence of overlapping text with the following previous publication(s), which needs to be addressed:

- https://doi.org/10.1016/j.nicl.2021.102685

In your revision ensure you cite all your sources (including your own works), and quote or rephrase any duplicated text outside the methods section. Further consideration is dependent on these concerns being addressed.

3) You indicated that you had ethical approval for your study. Please clarify whether minors (participants under the age of 18 years) were included in this study. If yes, in your Methods section, please ensure you have also stated whether you obtained consent from parents or guardians of the minors included in the study or whether the research ethics committee or IRB specifically waived the need for their consent."

4) In the ethics statement in the manuscript and in the online submission form, please provide additional information about the patient records/samples used in your retrospective study. Specifically, please ensure that you have discussed whether all data/samples were fully anonymized before you accessed them and/or whether the IRB or ethics committee waived the requirement for informed consent. If patients provided informed written consent to have data/samples from their medical records used in research, please include this information.

   "This study was granted by Samsung Medical Center Grant (OTC1190671)."

    "The authors declare that the research was conducted in the absence of any commercial or financial relationships that could be construed as a potential conflict of interest."

Reviewers' comments:

Reviewer's Responses to Questions

**Comments to the Author**

1. Is the manuscript technically sound, and do the data support the conclusions?

Reviewer #1: Yes

Reviewer #2: Yes

Reviewer #3: Partly

Reviewer #4: Partly

2. Has the statistical analysis been performed appropriately and rigorously? 

Reviewer #1: Yes

Reviewer #2: Yes

Reviewer #3: No

Reviewer #4: I Don't Know

3. Have the authors made all data underlying the findings in their manuscript fully available?

Reviewer #1: Yes

Reviewer #2: No

Reviewer #3: Yes

Reviewer #4: Yes

4. Is the manuscript presented in an intelligible fashion and written in standard English?

Reviewer #1: Yes

Reviewer #2: Yes

Reviewer #3: No

Reviewer #4: No

5. Review Comments to the Author

Reviewer #1: It’s a retrospective clinical study that needs a major revision.

1) Among all 1167 epilepsy surgery patients, 675 patients underwent one-step resection without invasive EEG studies. These 675 patients necessarily included patients with lateral temporal lobe epilepsy. While these 675 patients were excluded from the investigation target, i.e. the 63 lateral temporal lobe epilepsy, which means the conclusion ratio of 71.4% (45/63) seizure free could be inaccurate.

2) The author didn’t mention the exclusion criteria of the study. Since some lateral temporal lobe epilepsy with specific etiologies like tumor and AVM underwent invasive 2-step invasive EEG studies, what kinds of patients should undergo 1-step non-invasive study resection?

3) Since the solitary seizure-free rate of 71.4% (45/63) is not accurate, the author may compare the seizure-free rates between the lateral temporal lobe epilepsy and the mesial temporal lobe epilepsy among all invasive study temporal lobe epilepsy cases (131).

4) The author didn’t clarify the detail of the invasive EEG investigation. Did he implant deep electrodes into the hippocampus in all lateral temporal lobe epilepsy patients? Is it reasonable to do so in epilepsy with a clear lesion?

5) The author mentioned that he did anterior temporal lobectomy (ATL) and amygdalohippocampectomy (AH) in some lateral temporal lobe epilepsy patients. So how could he compare LTLE with MTLE since he did the same surgeries for them?

The author should revise the manuscript according to these questions before being considered for publication.

Reviewer #2: This is an interesting retrospective cohort study carried out between 1995 and 2018, among patients with lateral temporal lobe epilepsy (LTLE) who underwent resective surgery in a university-affiliated hospital of South Korea. Sixty-four patients were included in the study. The main results were: 1) statistically significant prognostic factors for postsurgical outcomes

were the duration of epilepsy before surgery and focal cortical dysplasia on postoperative histopathology at the 5-year follow-up; 2) cortical thinning was found in ipsilateral cingulate gyrus and contralateral parietal lobe in poor surgical group compared to good surgical group. The Authors also propose a model for predicting seizure outcomes 5 years after surgery using a nomogram.

As a minor observation (at page IX, lines 200-201) the Authors state that "there was no difference between the groups, except for the preoperative disease duration" Anyway, reading Table 1, also follow-up after surgery resulted statistically different in the two groups. So, the Authors should correct this sentence.

There are several typos in the text. For example at page XVIII, line 378, "...better tools are need..." should be changed in " ...better tools are needed..."; at the same page, line 384, "...the ppor surgical..." should be written "...the poor surgical...". The Authors should read agian their manuscript in order to correct typos.

Reviewer #3: This research aimed to evaluate the long-term surgical outcomes and identify possible prognostic factors in patients with lateral temporal lobe epilepsy. Retrospective analysis of sixty-four patients showed FCD was the ignificant prognostic factors for postsurgical outcomes at the 5-year follow-up. Cortical thinning was observed in ipsilateral cingulate gyrus and contralateral parietal lobe in poor surgical group compared to good surgical group. This study lacked of innovation and interesting. It seems that meaningful conclusions cannot be drawn from the results of this study

I have the following opinions：

1. In the part of abstract, the author should provide a more detailed method such as MRI analysis and clarify conclusions from the results.

2. In the part of cortical thickness, the author may add to MRI-negative image with same age as another control group.

3. Based on this study, I think the researchers used voxel based morphology to analyze cortical thickness. If so, voxel-wise FWE correction and threshold with FDR corrected should be used.

4. A significant English editing is necessary.

Reviewer #4: This is a retrospective, single center cohort study aiming to evaluate the short-term and long-term seizure outcomes following resective surgery in patients with lateral temporal lobe epilepsy. This study focused on interesting and important issue in resective epilepsy surgery. However, several limitations unfortunately dampen the enthusiasm for this study.

1. The definition of lateral temporal lobe epilepsy is unclear. It is based on electro-clinical findings? Imaging findings? The authors should clarify.

2. How many parents were underwent invasive EEG diagnostic with subdural or depth electrodes? Were these findings supported by clinical or imaging diagnosis?

3. It is remarkable, that in this study, the long-term seizure outcome is improved compared to the short-term outcome. What is the reason for this observation? The authors should discuss it more extensively.

4. Since the authors found FCD is significant factor for unfavorable outcome, it is important to know what kind of FCD (classification) was predominantly detected in this cohort.

5. What is the reason for worse outcome in patients with FCD in this cohort? The authors should explain this noteworthy observation

6. The subgroup analysis indicates patients with mesial abnormalities and co-existing FCD. This observation is presuming for including of patients with dual pathologies in this study. This could bias both, the findings and the conclusion made by the authors.

6. PLOS authors have the option to publish the peer review history of their article (what does this mean?). If published, this will include your full peer review and any attached files.

Reviewer #1: No

Reviewer #2: No

Reviewer #3: No

Reviewer #4: No

---

## [Author Response · Author response to Decision Letter 0]

15 May 2023

PONE-D-23-07317

Identifying Important Factors for Successful Surgery in Patients with Lateral Temporal Lobe Epilepsy

PLOS ONE

Author list: Jae Rim Kim, Hyunjin Jo, Boram Park, Yu Hyun Park, Yeon Hak Chung, Young-Min Shon, Dae-Won Seo, Seung Bong Hong, Seung-Chyul Hong, Sang Won Seo*, Eun Yeon Joo

May 2023

Dear editorial staff at the PLOS ONE

On behalf of the authors, I wish to express our appreciation of the reviewers' careful and helpful review of our manuscript. 

We revised the manuscript according to the reviewers’ suggestions, and we believe that these changes improved our manuscript. Please find enclosed response file addressing each of the reviewer's comments and the revised manuscript. The page and line references for the changes provided are based on the revised manuscript version. We hope that you and the reviewers now find the revised paper suitable for publication. 

Thank you for considering our manuscript for publication in PLOS one. We look forward to hearing from you soon.

Best Regards,

Jae Rim Kim

Response) According to PLOS ONE’s style requirements, we checked them over and revised. 

2. 1)Please include a complete copy of PLOS’ questionnaire on inclusivity in global research in your revised manuscript. Our policy for research in this area aims to improve transparency in the reporting of research performed outside of researchers’ own country or community. The policy applies to researchers who have travelled to a different country to conduct research, research with Indigenous populations or their lands, and research on cultural artefacts. The questionnaire can also be requested at the journal’s discretion for any other submissions, even if these conditions are not met. Please find more information on the policy and a link to download a blank copy of the questionnaire here: https://journals.plos.org/plosone/s/best-practices-in-research-reporting. Please upload a completed version of your questionnaire as Supporting Information when you resubmit your manuscript.

Response) According to PLOS ONE’s policy, we completed a questionnaire and uploaded as a S1 checklist. 

2) We noticed you have some minor occurrence of overlapping text with the following previous publication(s), which needs to be addressed:

- https://doi.org/10.1016/j.nicl.2021.102685

In your revision ensure you cite all your sources (including your own works), and quote or rephrase any duplicated text outside the methods section. Further consideration is dependent on these concerns being addressed.

Response) The same author in that previous publication was participated our study for cortical thickness measurement. We deleted that sub-section and moved to other sub-section. And we mentioned that the same method was used for analysis. 

3) You indicated that you had ethical approval for your study. Please clarify whether minors (participants under the age of 18 years) were included in this study. If yes, in your Methods section, please ensure you have also stated whether you obtained consent from parents or guardians of the minors included in the study or whether the research ethics committee or IRB specifically waived the need for their consent."

Response) We revised our methods section. 

(page iv, line 71-74)

Informed written consent was obtained from all participants when they admitted to epilepsy monitoring units (EMU), and it was obtained from the next of kin, caretakers, or guardians on behalf of the minors/children participants involved in this study. The Institutional Review Board of Samsung Medical Center approved the study protocol (IRB number 2021-03-068).

4) In the ethics statement in the manuscript and in the online submission form, please provide additional information about the patient records/samples used in your retrospective study. Specifically, please ensure that you have discussed whether all data/samples were fully anonymized before you accessed them and/or whether the IRB or ethics committee waived the requirement for informed consent. If patients provided informed written consent to have data/samples from their medical records used in research, please include this information.

Response) According to your recommendations, we added some sentences.

(page iv, line 74-75)

We had not access to information that could identify individual participants during study.

 "This study was granted by Samsung Medical Center Grant (OTC1190671)."

Response) We state that the funders had no role in study design, data collection and analysis, decision to publish, or preparation of the manuscript.

 "The authors declare that the research was conducted in the absence of any commercial or financial relationships that could be construed as a potential conflict of interest."

Response) There are no restrictions on sharing of data and/or materials of our manuscript. The authors declare that the research was conducted in the absence of any commercial or financial relationships that could be construed as a potential conflict of interest. This does not alter our adherence to PLOS ONE policies on sharing data and materials.

Response) According to your comment, we uploaded out study’s supplementary data.

Reviewers' comments:

Reviewer's Responses to Questions

Comments to the Author

1. Is the manuscript technically sound, and do the data support the conclusions?

Reviewer #1: Yes

Reviewer #2: Yes

Reviewer #3: Partly

Reviewer #4: Partly

2. Has the statistical analysis been performed appropriately and rigorously?

Reviewer #1: Yes

Reviewer #2: Yes

Reviewer #3: No

Reviewer #4: I Don't Know

3. Have the authors made all data underlying the findings in their manuscript fully available?

Reviewer #1: Yes

Reviewer #2: No

Reviewer #3: Yes

Reviewer #4: Yes

4. Is the manuscript presented in an intelligible fashion and written in standard English?

Reviewer #1: Yes

Reviewer #2: Yes

Reviewer #3: No

Reviewer #4: No

5. Review Comments to the Author

Reviewer #1: It’s a retrospective clinical study that needs a major revision.

1) Among all 1167 epilepsy surgery patients, 675 patients underwent one-step resection without invasive EEG studies. These 675 patients necessarily included patients with lateral temporal lobe epilepsy. While these 675 patients were excluded from the investigation target, i.e. the 63 lateral temporal lobe epilepsy, which means the conclusion ratio of 71.4% (45/63) seizure free could be inaccurate.

2) The author didn’t mention the exclusion criteria of the study. Since some lateral temporal lobe epilepsy with specific etiologies like tumor and AVM underwent invasive 2-step invasive EEG studies, what kinds of patients should undergo 1-step non-invasive study resection?

3) Since the solitary seizure-free rate of 71.4% (45/63) is not accurate, the author may compare the seizure-free rates between the lateral temporal lobe epilepsy and the mesial temporal lobe epilepsy among all invasive study temporal lobe epilepsy cases (131).

Response for 1-3) Thank you for your comments. We routinely conduct two-step invasive studies except typical patients with mesial temporal lobe epilepsy. Although there are specific lesions in lateral temporal area on MRI, we performed invasive monitoring to find ictal onset zone for tailored resection and minimize risk of surgical failures. We understand your concern, but we are sure include all LTLE patients with inclusion criteria. We excluded patients visited outpatient clinics less than 2 years.

4) The author didn’t clarify the detail of the invasive EEG investigation. Did he implant deep electrodes into the hippocampus in all lateral temporal lobe epilepsy patients? Is it reasonable to do so in epilepsy with a clear lesion?

Response) 

Thank you for your comment. We did not implant depth electrodes into hippocampus in all participants. We did it, if other presurgical studies such as functional imaging suggested impairments or alterations in mesial temporal structure. In those cases, we implanted depth to exclude mesio-lateral or mesial TLE and confirmed resection extent. We revised our method section according to your comment.

(page vi, line 134-136) 

All participants who were suspected LTLE underwent the invasive monitoring using a combination of subdural grids/strips with or without depth electrodes to confirm the ictal onset zone and determine the extent of resection.

5) The author mentioned that he did anterior temporal lobectomy (ATL) and amygdalohippocampectomy (AH) in some lateral temporal lobe epilepsy patients. So how could he compare LTLE with MTLE since he did the same surgeries for them?

The author should revise the manuscript according to these questions before being considered for publication.

Response)

Thank you for your comment. We did tailored resection of lateral temporal area in all patients and additional ATL and AH in some cases such as combined alterations in imaging. In other words, patients with MTLE underwent standard ATL and AH, but all patients with LTLE underwent tailored corticectomy or lesionectomy. But patients in the subgroup underwent AH simultaneously. So, surgical procedures were not same in LTLE compared to MTLE. 

Reviewer #2: This is an interesting retrospective cohort study carried out between 1995 and 2018, among patients with lateral temporal lobe epilepsy (LTLE) who underwent resective surgery in a university-affiliated hospital of South Korea. Sixty-four patients were included in the study. The main results were: 1) statistically significant prognostic factors for postsurgical outcomes

were the duration of epilepsy before surgery and focal cortical dysplasia on postoperative histopathology at the 5-year follow-up; 2) cortical thinning was found in ipsilateral cingulate gyrus and contralateral parietal lobe in poor surgical group compared to good surgical group. The Authors also propose a model for predicting seizure outcomes 5 years after surgery using a nomogram.

As a minor observation (at page IX, lines 200-201) the Authors state that "there was no difference between the groups, except for the preoperative disease duration" Anyway, reading Table 1, also follow-up after surgery resulted statistically different in the two groups. So, the Authors should correct this sentence.

There are several typos in the text. For example at page XVIII, line 378, "...better tools are need..." should be changed in " ...better tools are needed..."; at the same page, line 384, "...the ppor surgical..." should be written "...the poor surgical...". The Authors should read agian their manuscript in order to correct typos.

Response) Thank you for your comments. According to your recommendations, we revised some sentences including them. 

Reviewer #3: This research aimed to evaluate the long-term surgical outcomes and identify possible prognostic factors in patients with lateral temporal lobe epilepsy. Retrospective analysis of sixty-four patients showed FCD was the significant prognostic factors for postsurgical outcomes at the 5-year follow-up. Cortical thinning was observed in ipsilateral cingulate gyrus and contralateral parietal lobe in poor surgical group compared to good surgical group. This study lacked of innovation and interesting. It seems that meaningful conclusions cannot be drawn from the results of this study

I have the following opinions：

1. In the part of abstract, the author should provide a more detailed method such as MRI analysis and clarify conclusions from the results.

Response) Thank you for comments. We revised the abstract. 

(page ii, line 30-34)

Methods: This retrospective cohort study was conducted between January 1995 and December 2018 among patients who underwent resective surgery in a university-affiliated hospital. Patients were classified as LTLE if ictal onset zone was in lateral temporal area. Surgical outcomes were evaluated at 2 and 5 years. We subdivided based on outcomes and compared clinical and neuroimaging data including cortical thickness between two groups.

(page ii, line 46-48)

Conclusions: The identified predictors of unfavorable surgical outcomes may help in selecting optimal candidates and identifying the optimal timing for surgery among patients with LTLE. Additionally, cortical thinning was more extensive in the poor surgical group.

2. In the part of cortical thickness, the author may add to MRI-negative image with same age as another control group.

Response) Thank you for comments. We also hoped to analyze cortical thickness compared to healthy controls, but there was no available MRI data. These are our limitations and described in discussion. 

(page xx, line 439-440)

Fourthly, the absence of healthy controls in the cortical thickness analysis is another limitation.

3. Based on this study, I think the researchers used voxel based morphology to analyze cortical thickness. If so, voxel-wise FWE correction and threshold with FDR corrected should be used.

Response) Thank you for comments. We used voxel-wise FEW correction and threshold with FDR corrected at first, but we could not get meaningful results. Although the analysis was not ideal, we still believe that we would get significant results in larger participants. 

4. A significant English editing is necessary.

Response) Thank you for comments. We already got English editing service, but we checked our manuscript over in general and revised some parts.

Reviewer #4: This is a retrospective, single center cohort study aiming to evaluate the short-term and long-term seizure outcomes following resective surgery in patients with lateral temporal lobe epilepsy. This study focused on interesting and important issue in resective epilepsy surgery. However, several limitations unfortunately dampen the enthusiasm for this study.

1. The definition of lateral temporal lobe epilepsy is unclear. It is based on electro-clinical findings? Imaging findings? The authors should clarify.

Response) Thank you for comments. We defined LTLE based on the invasive study. If the seizure onset zone was in lateral temporal area, not mesial temporal structures (hippocampus, amygdala or parahippocampal gyrus) and basal temporal area, we diagnosed as LTLE. We described in method section.

(page iv, line 89-92)

Patients were diagnosed with LTLE if the seizure onset zone was in the lateral temporal area, not in mesial temporal structures or basal temporal area based on an invasive study. The seizure onset zone was defined as any paroxysmal ictal pattern that was distinct from background activity with clinical symptoms.

2. How many parents were underwent invasive EEG diagnostic with subdural or depth electrodes? Were these findings supported by clinical or imaging diagnosis?

Response) Thank you for comments. All enrolled participants were underwent invasive study with subdural electrodes in some cases. We proceeded invasive monitoring, if clinical or imaging diagnosis suggested LTLE. Therefore, presurgical diagnostics supported LTLE and invasive study confirmed it. 

3. It is remarkable, that in this study, the long-term seizure outcome is improved compared to the short-term outcome. What is the reason for this observation? The authors should discuss it more extensively.

Response) Thank you for comments. According to your recommendations, we additionally described in our discussion. 

(page xvi, line 327-333)

Interestingly, our study showed surgical outcome improved over time. This may be related to study population. As reported by Tellez-Zenteno[24], long-term seizure freedom was highest in patients with tumor, but patients older than 50 years at time of surgery was lowest compared to short-term seizure freedom in temporal lobe surgery. We included patients with tumorous lesions and the average age of surgery was 29.0 years. There are also results that surgical outcomes improved patients with FCD[25], and it is agreement with our results.

4. Since the authors found FCD is significant factor for unfavorable outcome, it is important to know what kind of FCD (classification) was predominantly detected in this cohort.

Response) Thank you for the comment. We tried to report types of FCD, but it was tough. Because more than half of participants were performed epilepsy surgery before 2011, there was no available data of FCD classification. Among available data, 30% (3/10) of each was type 1 and 2 and remaining 40% (4/10) was type 3. So we thought interpretation of it might lead to bias due to limited data, we did not describe it in the manuscript.

5. What is the reason for worse outcome in patients with FCD in this cohort? The authors should explain this noteworthy observation

Response) Thank you for comments. According to your recommendations, we additionally described in our discussion. 

(page xvi-xvii, line 347-352)

However, the predictive value of FCD has been controversial, which could be attributed to differences in the study population [8, 27]. Some studies have suggested that complete resection, including dysplasia, is more predictable than FCD itself [28, 29]. This may be one of the reasons why FCD attributes to worse prognosis in our study. In patients with non-lesional MRI, resection margin was determined only by ictal onset zone. It might lead to incomplete resection and seizure recurrence.

6. The subgroup analysis indicates patients with mesial abnormalities and co-existing FCD. This observation is presuming for including of patients with dual pathologies in this study. This could bias both, the findings and the conclusion made by the authors.

Response)

Thank you for comments. Although there were MRI abnormalities in some patients, all ictal onset was seen in lateral temporal areas suggesting LTLE. So, we described reasons of signal changes as seizure related changes or kindling in our discussion. 

(page xix, line 408-419)

As confirmed by invasive monitoring, ictal onset zones were in the lateral temporal area in all patients including the subgroup. As a result, signal changes observed in the mesial temporal area could be attributed to other mechanisms. First, seizures can induce signal changes in MRI, which are transient abnormalities that occur after seizure activity in the epileptogenic zone or distinct regions connected by an epileptic network[43]. In some cases, MRI scans were performed after clustering seizures on the last day of EMU monitoring, which could impact the observed signal changes on MRI. Second, the kindling model might be related to these alterations. Kindling refers to a change in seizure characteristics and behaviors resulting from recurrent seizures due to focal electrical stimulation.[44] Limbic circuits, including the hippocampus and amygdala, are highly vulnerable to kindling, which can eventually lead to epileptogenicity in affected regions. One study showed changes observed in the mesial temporal area may be due to the kindling process.[45]

6. PLOS authors have the option to publish the peer review history of their article (what does this mean?). If published, this will include your full peer review and any attached files.

Do you want your identity to be public for this peer review? For information about this choice, including consent withdrawal, please see our Privacy Policy.

Reviewer #1: No

Reviewer #2: No

Reviewer #3: No

Reviewer #4: No

---

## [Decision Letter · Decision Letter 1]

19 Jun 2023

Identifying important factors for successful surgery in patients with lateral temporal lobe epilepsy

PONE-D-23-07317R1

Dear Dr. Joo,

We’re pleased to inform you that your manuscript has been judged scientifically suitable for publication and will be formally accepted for publication once it meets all outstanding technical requirements.

Kind regards,

Tommaso Martino, M.D.

Academic Editor

PLOS ONE

**Comments to the Author**

1. If the authors have adequately addressed your comments raised in a previous round of review and you feel that this manuscript is now acceptable for publication, you may indicate that here to bypass the “Comments to the Author” section, enter your conflict of interest statement in the “Confidential to Editor” section, and submit your "Accept" recommendation.

Reviewer #1: All comments have been addressed

Reviewer #2: All comments have been addressed

Reviewer #4: All comments have been addressed

2. Is the manuscript technically sound, and do the data support the conclusions?

Reviewer #1: Yes

Reviewer #2: Yes

Reviewer #4: Partly

3. Has the statistical analysis been performed appropriately and rigorously? 

Reviewer #1: Yes

Reviewer #2: Yes

Reviewer #4: I Don't Know

4. Have the authors made all data underlying the findings in their manuscript fully available?

Reviewer #1: Yes

Reviewer #2: Yes

Reviewer #4: Yes

5. Is the manuscript presented in an intelligible fashion and written in standard English?

Reviewer #1: Yes

Reviewer #2: Yes

Reviewer #4: No

6. Review Comments to the Author

Reviewer #1: All my comments have been addressed and revised by the authors. I have no additional comment for this paper.

Reviewer #2: The Authors addressed my personal queries and comments, and I think taht this paper is now improved.

Reviewer #4: (No Response)

7. PLOS authors have the option to publish the peer review history of their article (what does this mean?). If published, this will include your full peer review and any attached files.

Reviewer #1: No

Reviewer #2: No

Reviewer #4: No

---

## [Editor Report · Acceptance letter]

21 Jun 2023

PONE-D-23-07317R1 

Identifying important factors for successful surgery in patients with lateral temporal lobe epilepsy 

Dear Dr. Joo:

I'm pleased to inform you that your manuscript has been deemed suitable for publication in PLOS ONE. Congratulations! Your manuscript is now with our production department. 

Kind regards, 

on behalf of

Dr. Tommaso Martino 

Academic Editor

PLOS ONE